# In Silico and In Vitro Analysis of MAP3773c Protein from *Mycobacterium avium* subsp. *Paratuberculosis*

**DOI:** 10.3390/biology11081183

**Published:** 2022-08-06

**Authors:** Esteban Hernández-Guevara, José A. Gutiérrez-Pabello, Kaina Quintero-Chávez, Mirna Del Carmen Brito-Perea, Lilia Angélica Hurtado-Ayala, German Ibarra-Molina, Omar Cortez-Hernández, Dulce Liliana Dueñas-Mena, Ángela Fernández-Otal, María F. Fillat, Bertha Landeros-Sánchez

**Affiliations:** 1Facultad de Odontología, Universidad Autónoma de Baja California, Tijuana 22390, Mexico; 2Facultad de Medicina Veterinaria y Zootecnia, Universidad Nacional Autónoma de México, México City 04510, Mexico; 3Facultad de Ciencias Químicas e Ingeniería, Universidad Autónoma de Baja California, Tijuana 22390, Mexico; 4Departamento de Bioquímica y Biología Molecular y Celular, Facultad de Ciencias e Instituto de Biocomputación y Física de Sistemas Complejos, Universidad de Zaragoza, 50009 Zaragoza, Spain

**Keywords:** MAP3773c protein, docking models, in silico models

## Abstract

**Simple Summary:**

Paratuberculosis is a disease that is caused by *Mycobacterium avium* subsp. *paratuberculosis*, a bacterium that survives inside a cell to cause disease. This bacterium therefore needs the nutrients of the cell to survive. Zinc and iron are very important elements in its nutrition and are necessary to carry out many of its survival functions, so the cell develops mechanisms to eliminate these pathogenic bacteria to continue living. One of these mechanisms is the elimination of iron as a strategy to kill the bacteria. In this research, we took on the task of studying one of the proteins of the bacterium called MAP3773c, along with its structure, some of its properties and its particular characteristics. In relation to the affinity for zinc and iron to bind to it, we are interested in discovering and making it known to the scientific community whether MAP3773c is related to the pathology of the disease.

**Abstract:**

Paratuberculosis is a disease caused by *Mycobacterium avium* subsp. *paratuberculosis* (MAP). It is of great interest to better understand the proteins involved in the pathogenicity of this organism in order to be able to identify potential therapeutic targets and design new vaccines. The protein of interest–MAP3773c–was investigated, and molecular modeling in silico, docking, cloning, expression, purification, and partial characterization of the recombinant protein were achieved. In the in silico study, it was shown that MAP3773c of MAP has 34% sequence similarity with *Mycobacterium tuberculosis* (MTB) FurB, which is a zinc uptake regulator (Zur) protein. The docking data showed that MAP3773c exhibits two metal-binding sites. The presence of structural Zn^2+^ in the purified protein was confirmed by SDS-PAGE PAR staining. The purification showed one band that corresponded to a monomer, which was confirmed by liquid chromatography–mass spectrometry (LC-MS). The presence of a monomer was verified by analyzing the native protein structure through BN-SDS-PAGE (Native Blue (BN) Two-Dimensional Electrophoresis) and BN–Western blotting. The MAP3773c protein contains structural zinc. In conclusion, our results show that MAP3773c displays the features of a Fur-type protein with two metal-binding sites, one of them coordinating structural Zn^2+^.

## 1. Introduction

Paratuberculosis (PTB), which is also known as Johne’s disease, is a chronic granulomatous enteritis caused by *Mycobacterium avium* subsp. *Paratuberculosis* (MAP) [1]. In this disease, the fight for survival within the host cell depends on MAP’s ability to acquire nutrients and metals to carry out different functions, such as activation of the catalytic sites of many enzymes for survival, adaptation, and pathogenicity within the cell [2,3]. Iron and zinc are essential metals for all living organisms [4]. Therefore, during the process of infection, there is a battle between the pathogen and the host cell. For example, through nutritional immunity, the host depletes metals through the expression of factors that limit their availability and, consequently, limit the pathogen’s survival [5].

MAP can infect and survive in macrophages. Throughout its evolution, it has developed several defense mechanisms that are often related to the expression of genes controlled by the family of transcriptional regulators, which are involved in iron acquisition. The FURfamily proteins include diverse metal regulators, such as Fur (Fe^2+^), Zur (Zn^2+^), and Mur (Mn^2+^), as well as stress sensors of peroxide (PerR) and heme (Irr) [6]. These proteins also regulate the oxidative stress genes and virulence determinants of the pathogenic bacteria that are necessary to initiate the expression of virulence factors [7]. Usually, FUR proteins join a consensus region called the iron box [8]. In the presence of a metal cofactor, the structure dimerizes; however, it is also capable of multimerizing [9,10]. FUR dimers bind to the consensus sequence in target promoters, avoiding the binding of the RNA polymerase and consequently the transcription of the target gene [11]. In the majority of the bacteria, such as *Escherichia coli* K12 (*E. coli*) [12], *Vibrio cholerae* [13], *Microcystis aeruginosa* [14], *Pseudomonas aeruginosa* [15], *Helicobacter pylori* [16], *Neisseria meningitidis* [17], *Acinetobacter baumannii* [18], and *Listeria monocytogenes* [19], among others, the genes involved in the biosynthesis of siderophores and iron transport are regulated by the Fur paralogue.

It has been determined that in MAP, iron metabolism is controlled by *IdeR* (MAP2827). This *IdeR* region was identified by alignment with the *IdeR* region of *M. tuberculosis* (MTB) and showed 86% similarity. Computational predictions identified 24 genes that may be regulated by *IdeR* and others that are regulated in human macrophage cultures, among which *map3773c* was not found. The participation of the *IdeR* of MAP in the recognition and protection of the iron box in the promoters *mbtB* and *bfrA* in MAP has already been clearly determined. This demonstrated that the *IdeR* of MAP is a protein that binds to the promoter regions of these two genes that are regulated by iron and is likely to function as a transcriptional regulator in MAP [20]. On the other hand, in relation to zinc metabolism, a genomic island of 64 zinc-sensitive genes has been located; these were grouped in a single 90 kb locus within the genome, which includes the region from MAP3725 to MAP3788 [21]. An analysis revealed three regions that correspond to zinc uptake: the ZnuABC transporter (MAP 0487c-0489c) and two transporters located in ZnGI, which are *mptABC* (MAP3736c-3734c) and an ABC-type Mn^2+^/Zn^2+^ transporter (MAP3773c-3776c). All were induced by zinc starvation. The ABC-type Mn^2+^/Zn^2+^ transporters and the *mptABC* transporter do not have homologues in other mycobacteria, possibly including MAP, allowing them to avoid zinc deficiency more efficiently [21,22]. The *map3773c* gene is located on the genomic island LSP P^15^, which was acquired by MAP through horizontal gene transfer, which possibly encodes an alternative iron absorption system [22]. Recently, the MAP3773c protein of MAP was characterized and analyzed by CHiP-seq, in vivo, in silico, and in vitro by EMSA. In this research, three peaks of ChiP-seq that were associated with the regulation of iron were identified–namely MAP3638c under conditions without iron, MAP3736c under iron-rich conditions, and MAP3737, which is the iron sensor that activates a hypothetical master regulator (MR). It was determined that MAP3773c is a Fur protein that regulates the expression of genes involved in the absorption and storage of iron in culture conditions with high iron concentrations and with poor iron conditions [23]. The objective of this work was to partially characterize the gene product MAP3773c in silico and by docking analysis, as well as to perform cloning, expression purification, and biochemical studies, which are necessary for subsequent studies of this protein.

## 2. Materials and Methods

### 2.1. Model-Based Prediction of the Molecular Structure of MAP3773c

The structure of the MAP3773c protein with the accession number AAS06323.1 was predicted using the FASTA format information on the National Center for Biotechnology Information (NCBI, https://www.ncbi.nlm.nih.gov/genbank/) (accessed on 21 January 2014) website. The MAP3773c sequence of MAP in FASTA format was introduced to JPRED 4 Jnet version: 2.3.1 James Cuff and Jonathan Barber, United Kingdom, 2015 online software [24], a server for predicting the secondary structure of proteins. We then conducted an identification search in Chimera 1.12 (UCSF Chimera), a visualization system for exploratory research and analysis, into which we loaded the amino acid sequences of interest, including the crystal contacts from PDB IDs: 2O03, PDB 3MWM, PDB ID: 4RB1 and PDB ID: 5FD6 proteins. The alignment was performed using the algorithm with a BLOSUM62 matrix. The homology-based model was made using Chimera software; this included comparative modeling by MODELLER [25]. To obtain the generated models, we created an alignment for the root mean square deviation (RMSD). After the completion of the model and the comparison, we proceeded to review the amino acid residues (one by one) to compare them with the information given by JPRED 4. After verifying the positions of the amino acid residues, we proceeded to decrease the contacts that emerged between the side chains; this was carried out with the Vega ZZ 3.1.1.42 Molecular Modeling Toolkit [26,27]. After finishing the analysis, figures were generated using Chimera UCSF and VDM. Finally, the sequences of the proteins with similarity were used to perform an alignment using Geneious 7.1.3 Biomatters Ltd., Auckland, New Zealand software (https://www.geneious.com) (accessed on 20 January 2021). To determine whether MAP3773c had a structural interaction with metals, we performed an in silico docking analysis to determine if it has an affinity to bind with Zn^2+^ and Mn^2+^. Specifically, this was carried out to obtain more information about the protein and to help us determine its function and characteristics. This in silico docking analysis was performed online on a server called MIB, which shows the interactions of amino acids with the metal in different models or templates of the protein and the metal. Each template receives a score from that model within 3.5 Å of the center of the metal ion. The method uses the fragment transformation technique, performing a local structure comparison between the metal ion binding residue templates and a query protein structure with unknown binding sites. After the comparison, each query protein residue is assigned an alignment score using a scoring function composed of two criteria used to assess how well the sequence and structure are conserved. By setting a threshold for the alignment scores, residues exceeding the threshold are expected to be metal-binding residues [28].

### 2.2. Cloning and Expression of map3773c

In this study, we conducted the cloning, expression, purification, and a partial functional analysis of the MAP3773c protein (NCBI Reference Sequence NC_02944.2). The gene *map3773c* was amplified from DNA of the ATCC strain of MAP 19698 using a Taq polymerase (Life Technologies, Carlsbad, CA, USA), and the corresponding sense and antisense oligonucleotides (GAGCTCGTGTCATCGCCCGCTGGG and AAGCTTTCACGGTTGTGTGTTTTG, respectively) (PCR3.1) were used. The reaction conditions were a denaturation cycle of 94 °C for 10 min, followed by 30 cycles of 94 °C denaturation for 30 s, annealing at 63 °C for 30 s and extension at 72 °C for 30 s, then a final cycle at 72 °C for 10 min. The DNA was separated by 1.2% agarose electrophoresis and stained with ethidium bromide to determine the size of the DNA fragments, using a molecular weight marker as a reference. The amplified region of MAP3773c was 420 bp from the start to the stop codon. This amplified fragment of *map3773c* was cloned in TOPO PCR 2.1 (Invitrogen, Waltham, MA, USA) and subcloned in the expression vector pRSET-A (Invitrogen). The cloning product of pCR2.1TOPO-*map37773c* was used to transform competent *E. coli* DH5α cells, and the fragment of *map3773c* was obtained using the restriction enzymes *SacI* and *HindIII*. The cloning was confirmed by sequencing (Eton Bioscience, San Diego, CA, USA). This fragment was subcloned in pRSET-A to create a protein fused with a 6-His label at the N terminal. To subsequently confirm the site of cloning and the integrity of *map3773c*, sequencing of the plasmid containing the *map3773c* gene was performed.

To analyze the expression of *map3773c*, the expression vector pRSET-*map3773c* was introduced into competent BL21 (DE3) (Invitrogen) cells for transformation by thermal shock. The clones obtained were selected and inoculated in 10 mL of a Luria–Bertani (LB, HiMedia Laboratories Pvt. Ltd., Mumbai, India) medium with 50 μg/mL ampicillin to verify that they contained the plasmid of interest. The culture was incubated for 12 h at 37 °C with constant agitation at 200 rpm. To express the gene of the MAP3773c protein, 10 mL of the previous culture was added into 1 L of LB under the same conditions. When the optical density reached an absorbance in the range of 0.4–0.6 nm, isopropyl β-d-1-thiogalactopyranoside (IPTG, Promega, Madison, WI, USA) was added to a final concentration of 1.0 mM, and the mixture continued incubating for 16 h at 30 °C and 200 rpm. The theoretical molecular weight was obtained from an online server Quest CalculateTM Peptide and Protein Molecular Weight Calculator. AAT.Bioquest (https://www.aatbio.com/tools/calculate-peptide-and-protein-molecular-weight-mw, accessed on 15 June 2022) [29].

### 2.3. Protein Purification Conditions and Metal Oligomerization

The expression level was verified, and the culture was centrifuged at 20,000 rpm at 4 °C for 30 min. MAP3773c was purified according to [30], with modifications. In a single step, 10 g of cells was resuspended in 50 mL of Buffer A (0.1 M NaH_2_PO_4_, 0.01 Tris, 2 M guanidine—HCl, pH 8) with 1 mM of phenylmethylsulfonyl fluoride (PMSF, Roche), then placed in an ice bath and lysed with an ultrasonicator (Dr. Hielscher, 2005) with pulses of 190 W lasting 45 s. The clarified lysates were obtained by centrifuging at 48,000× *g* for 30 min. The resulting supernatant was incorporated into a column containing the Chelating Sepharose Fast Flow (GE Healthcare, Chicago, IL, USA) resin, previously immobilized in 0.25 M ZnSO_4_, following the manufacturer’s instructions. The volume of supernatant collected from the column was called the dead extract. The column was washed with five volumes of 0.5 M (NH_4_)_2_SO_4_ in Buffer A, then washed a second time with another five volumes of 35 mM glycine in Buffer A. The absorbance was measured at 280 nm, followed by continual washing with this last buffer until the absorbance was 0.1 nm. MAP3773c was then eluted with a gradient of imidazole to 1 M in Buffer A, and the volumes were collected in aliquots of 1 mL. Aliquots at a ratio of 15 μL and 85 μL of bio-distilled water were treated with 100 μL of 10% trichloroacetic acid (TCA) in 2 mL tubes, which were incubated for 20 min on ice and then centrifuged at 10,000 rpm for 10 min. The supernatant was removed and 100 μL of absolute ethanol was added, then the mixture was centrifuged, and the supernatant was discarded. The samples were dried for 10 min at 37 °C to eliminate ethanol, then diluted with 15 μL of an acetate buffer at pH 4. Subsequently, the purified protein was analyzed in a polyacrylamide SDS-PAGE gel at 17% (*w*/*w*) according to [31] and stained with Coomassie Brilliant Blue. The aliquots where the protein was found were concentrated using 10 kDa Centricon (Millipore, Burlington, MA, USA) membrane tubes. The guanidinium chloride and imidazole were then eliminated through dialysis in a buffer of 10 mM acetic acid and acetate at pH 4. The protein concentration was quantified by the Bradford method by measuring the absorbance at 280 nm, using the theoretical extinction coefficient of the protein supplied by ProtParam via the ExPASy server (http//:web.expasy.org/protparam) (accessed on 30 January 2016) for the oxidized protein (ε 280 nm = 4720 nm mM^−1^ cm^−1^). The samples were then stored in aliquots of 1 mg/mL at −20 °C until use. The isoelectric point (IP) was determined with the ProtParam program of ExPASy (http://:web.expasy.org/protparam, accessed on 30 January 2016) after introducing the amino acid sequence of the recombinant MAP3773c protein. For the oligomerization experiments with metals such as Zn^2+^ and Mn^2+^, we performed the experiments with MAP3773c under reducing and non-reducing conditions, with and without β-mercaptoethanol, with DTT at a final concentration of 10 mM with and without β-mercaptoethanol, with Mn^2+^ at a final concentration of 15 mM with and without β-mercaptoethanol, and with Zn^2+^ to final concentration of 15 mM with and without β-mercaptoethanol.

### 2.4. Circular Dichroism (CD)

The MAP3773c protein was also analyzed by spectropolarimetry at a range of 195–265 nm to determine the circular dichroism characteristics and elucidate the secondary structure; the equipment used was a Chirascan spectropolarimeter (Applied Photophysics Ltd., Leatherhead, UK) Surrey, Union Kingdom. The tests were performed at a temperature of 25 °C for 12 s per point, with a trajectory length of 1.0 cm and a length of passage and bandwidth of 1.0 nm. The protein was tested at a concentration of 30 μM under three different conditions: protein in a buffer of 10 mM acetate and acetic acid (pH 5.5). The data were generated on the DichroWeb server “Online analysis for Protein Circular Dichroism spectra” (http://dichroweb.cryst.bbk.ac.uk/html/home.shtml, accessed on 10 July 2016) [32,33] using the analytical software of the electronic site CDSSTR. The spectrum results were expressed in units of ∆ε (the molar absorbance unit for circular dichroism residues measured in M^−1^ cm^−1^). The experimental results were contrasted with the in silico results of the bioinformatics tool PSIPRED (http://bioinf.cs.ucl.ac.uk/psipred/, accessed on 12 July 2016 ) The Bloomsbury Center for Bioinformatics, University College London and Birkbeck University of London) [34].

### 2.5. Cross-Linking and Structural Detection of Zinc

Cross-linking experiments were performed on purified MAP3773c at a concentration of 30 µM in an acetate buffer (pH 4.0) with 0.5% glutaraldehyde to a final volume of 20 µL. The reaction mixture was incubated for 30 min in the dark at room temperature. Finally, the reaction was heated under denaturing conditions with 5% β-mercaptoethanol, and the resulting products were analyzed by SDS-PAGE gel at 12%. For the detection of the Zn^2+^ structural protein recombinant MAP3773c, we first conducted SDS-PAGE electrophoresis under four different conditions: MAP3773c protein under non-reducing and reducing conditions (a) with 5% β-mercaptoethanol, (b) with DTT at a final concentration of 10 mM, (c) without β-mercaptoethanol, and (d) with H_2_O_2_ without β-mercaptoethanol. Under all conditions, protein was dyed with 50 mL of a stain which contained a buffer of 20 mM Tris-HCl (pH 8.0), 100 mM NaCl, 5% *v*/*v* glycerol and 500 µM of 4-(2-pyridylazo)-resorcinol monosodium hydrated (PAR) dye from Sigma-Aldrich [35], mixed by gentle agitation for 20 min and then added to 500 µL of H_2_O_2_ to 30%. The PAR dye reacts with zinc ions to form a complex of orange that can be observed with the naked eye and disappears after 5–10 min. Later, the same PAR dye in the same gel was stained with Coomassie blue to check for the presence of proteins.

### 2.6. Western Blotting

We confirmed the identity and expression of the protein MAP3773c by Western blotting according to [36]. The protein was separated into a 12% SDS-PAGE gel [31] and calibrated using a prestained molecular weight marker (Novex, Life Technologies, Carlsbad, CA, USA) to determine the size of the protein. The purified protein was subjected to electroblotting on a membrane of polyvinylidene difluoride (Trans-Blot Turbo, Bio-Rad, Hercules, CA, USA) in a semiliquid condition for 6 h at 300 mA, and the membrane was subsequently blocked with 5% of Svelty milk in 1× PBS for 1 h at 37 °C, then washed. The first antibody used was an anti-His-tag recombinant rabbit polyclonal antibody (Life Technologies, Delhi, India) and the second antibody was a goat anti-rabbit IgG (H + L)-HRP conjugate (Bio-Rad, Hercules, CA, USA). The membrane was then stained with 3-amino-9-ethyl carbazole (Sigma-Aldrich, St. Louis, Mo, USA) and hydrogen peroxide for 15 min in the dark and was finally washed and dried. To determine whether MAP3773c had characteristics similar to those of FurA and FurB proteins, Western blotting was carried out with *Anabaena* sp. PCC7120 antibodies (kindly provided by Dr Maria F. Fillat’s laboratory).

### 2.7. BN-SDS PAGE and BN-PAGE Immunoblotting

To determine whether MAP3773c protein is found in complexes that adopt its native functional conformational state, we performed a two-dimensional electrophoresis. First, we separated them by their isoelectric point and later by their molecular weight. We carried out the experiments according [37], using 1.4 µg/µL of protein from an extract of *E. coli* BL21 (DE3) pRSET-*map3773c* bacteria, induced with IPTG at a final concentration of 1 mM, as described above. We carried out the experiments step by step as indicated in the protocol of [37] with the same solution contents and/or conditions; however, a bacterial pellet was also used. Electrophoresis was performed to stain the gel with Coomassie Blue, followed by further electrophoresis to perform immunoblotting, which was revealed by the anti-His-tag antibody under the same conditions used for the Western blotting experiment for the purified protein with same molecular weight, as described above.

### 2.8. Liquid Chromatography–Mass Spectrometry (LC-MS)

#### 2.8.1. Optimization of the Mass Spectrometer (External Calibration)

To ensure that the instrument worked within the specifications, the parameters were adjusted with a Calmix solution (N-butylamine, caffeine, Met-Arg-Phe-Ala (MRFA) and Ultramark 1621 (Pierce LTQ Veils ESI positive ion calibration solution)). These calibrators were used to calibrate the module’s LTQ veils with ion traps and the module’s Orbitrap with the FT (Fourier transform) mass detector in positive ESI ionization mode. The N-butylamine was used to extend the calibration of the masses to lower values of *m*/*z* (below 73.14 kDa) because our protein hypothetically has a mass of 20 kDa. This type of calibration allowed the determination of the molecular masses with variations of accuracy below 5 ppm (parts/million).

#### 2.8.2. Spectrometric Analysis

The MAP3773c sample, which had previously been desalted with ZipTip C18 (Millipore; Billerica, MA, USA), was injected into an LC-MS (liquid chromatography–mass spectrometry) system composed of a ACCELA (Thermo-Fisher Co.; San Jose, CA, USA) pump coupled to a mass spectrometer with LTQ-Orbitrap veils (Thermo-Fisher Co., San Jose, CA, USA) and a nanoelectrospray-type ionization source (ESI). In the chromatography of the nanoflux fluids online, an isocratic system of 50–50% (water/acetonitrile with 0.1% formic acid) was used for 20 min with a homemade capillary needle. The flow rate of the LC system was 300 nanoliters/minute.

#### 2.8.3. MALDI-TOF MS, (Matrix-Assisted Laser Desorption/Ionization Time of Flight) MS

The MALDI-TOF MS was carried out in an external service at the National Laboratory for the Structure of Macromolecules LANEM-IQ-UNAM, Mexico City.

The purified protein was dialyzed against sodium acetate (membrane 3.5 kDa) and the mass was determined using (MALDI-TOF) mass spectrometry (Microflex; Bruker Scientific LLC, Billerica, MA, USA). The matrix used was a saturated solution of sinapinic acid in 30% (*v*/*v*) aqueous acetonitrile, 0.1% (*v*/*v*) trifluoroacetic acid, and the sample was calibrated with BSA. The protein concentration was 4.6 mg/mL.

## 3. Results and Discussion

### 3.1. Homology of MAP3773c to Other Proteins of the FUR Family

The alignment was carried out with 36 proteins that showed high similarity to *MAP3773c*. To carry out the study of homology modeling, we selected the structures of the four proteins and displayed the secondary structure with the greatest similarity according to the alignment provided by the JPRED 4.0 program. These proteins were Fur B from *M.tuberculosis* (PBD ID: 2O03), Zur from *Streptomyces coelicolor* (PDB ID: 3MWM), Fur-Mn^2+^ from *Magnetospirillum gryphiswaldense* MSR-1 v2 (PDB ID: 4RB1), and Mur from *Rhizobium leguminosarum* bv. viciae (PDB ID: 5FD6).

Figure 1 presents the alignments and conserved domain protein, made online in the database of the NCBI (https://www.ncbi.nlm.nih.gov/Structure/cdd/cddsrv.cgi, accessed on 16 September 2021) [38]. Figure 2 presents the in silico model of MAP3773c compared with the crystalline structures of several proteins (Fur B from *M.tuberculosis* (PBD ID: 2O03), Zur from *Streptomyces coelicolor* (PDB ID: 3MWM), Fur-Mn^2+^ from *Magnetospirillum gryphiswaldense* MSR-1 v2 (PDB ID: 4RB1), and Mur from *Rhizobium leguminosarum* bv. viciae (PDB ID: 5FD6). The crystalline structure of FurB of *M. tuberculosis* (PBD ID: 2O03) in Figure 2a, compared with MAP3773c, presented as a monomer showing 34% identity with an RMSD of 1 Å. PDB ID: 2O03 exhibits the function of a Zur protein. Its structure consists of a chain including a domain of dimerization, a domain of binding to DNA, and three sites for binding to Zn [39], and this Zn-dependent protein showed close identity with MAP3773c. Another of the proteins with which MAP3773c had high homology was the PDB ID: 3MWM (Figure 2b) of *Streptomyces coelicolor* (strain ATCC BAA-471/A3 (2)/(M145)), which showed an identity of 33%, only four gaps, and an RMSD of 1 Å; this structure also contains three Zn binding sites per monomer, and the protein works as a Zur-type protein [40]. Another of the proteins was the PDB ID: 4RB1of *Magnetospirillum gryphiswaldense* MSR-1v2 Fur—Mn^2+^ [41] (Figure 2c), which had 26% identity with MAP3773c and a protein crystal structure in dimeric form with a DNA interaction in the Fur box and 2.75 Å resolution. The last model was made with the protein PDB ID: 5FD6 from *Rhizobium leguminosarum* bv. *viciae* (Figure 2d), a zinc-bound manganese uptake regulator with a resolution of 2.4 Å, only four gaps [42], and 23% identity with the MAP3773c protein.

The superposition of the MAP3773c model with the structures of proteins such as PDB IDs: 2O03, 3MWM, 4RB1, and 5FD6 suggests that MAP3773c interacts with a structural zinc atom in the cysteines (91, 94, 131, and 134). If we consider the intermolecular distances of the different superpositions, the structure that showed the lowest atomic distance was from PDB ID: 2O03, with the exception of cysteine 94, which presented an intermolecular distance of 2.18 Å, with the other crystals (PDB IDs: 3MWM, 4RB1, and 5FD6) having an intermolecular distance of 1.77 Å. We suggest that MAP3773c possibly has another Zn^2+^ atom at the histidine (85, 87, and 123) and glutamic acid (107) positions, according to the data from the overlays with the PDB ID: 2O03 crystal, which is linked to the amino acids histidine (87,89 and 125) and glutamic acid (108). In the case of the PDB ID: 3MWM crystal, it has a zinc atom at the histidine positions (84, 86, and 122) and glutamic acid (105). On the other hand, the PDB ID: 5FD6 crystal has histidine (36, 91, and 93) and glutamic acid 84. However, it is also possible that MAP3773c forms a bond with Mn^2+^ at the histidine positions (85, 87, and 123) and glutamic acid (107), according to the superposition with the PDB ID: 4RB1 crystal, which forms the bond with the Mn^2+^ with histidine (87 and 125), aspartic acid (89), glutamic acid (108), and water at position 301. The aspartic acid at position 89 of the PDB ID: 4RB1 crystal is not present in MAP3773c; instead, it has one at position 72 and a glutamic acid at position 82, which probably has an interaction with the Mn^2+^. The intermolecular distances derived from the superposition of the crystals of PDB ID: 2O03, were 1.0, 2.43, and 2.34 Å of histidine (87, 89, and 125) and 3.11 Å of glutamic acid (108). In the case of the PDB ID: 3MWM overlay, the histidine (84, 86, and 122) was at 1.96, 2.05, and 2.04 Å, respectively, and the glutamic acid (105) was at 1.64 Å. In PDB ID: 4RB1, histidine (87 and 125) was at 2.31 and 2.07 Å, respectively, aspartic acid (89) was at 2.05 Å, glutamic acid (108) was at 2.41 Å, and water (301) was at 2.36 Å. In the case of PDB ID: 5FD6, the intermolecular distances of the overlay were 2.23, 2.28, and 2.31 Å for histidine (36, 91, and 93), respectively, and 2.18 Å of glutamic acid (84). The intermolecular distances in the overlay with PDB ID: 3MWM were smaller with respect to those of the PDB ID: 2O03 crystal, with the exception of histidine (87) in PDB ID: 2O03, which was smaller; however, the amino acids with which these interactions with structural zinc were formed were found in the MAP3773c model, in positions very close to these two crystals. In PDB ID: 4RB1, not all the amino acids with which Mn^2+^ interacts were found in our model. A similar case was found for the PDB ID: 5FD6 crystal: The histidine in position 36 was not located in MAP3773c, the remaining two histidines were found in positions 91 and 93, and glutamic acid was found at position 84; in the case of our model, the histidines were in positions 85 and 87, and glutamic acid was found at 84. In the case of the overlay with PDB ID: 3MWM, the distances were smaller, so we think it is most likely that MAP3773c contains zinc instead of manganese in these positions and that the shape of the fold is more similar to that of PDB ID: 3MWM crystal, a question that will be elucidated when we obtain the crystal.

### 3.2. Inferring Metal Binding Sites (Zn^2+^ and Mn^2+^) in MAP3773c Protein 

In order to determine if the MAP3773c protein has an affinity for metals, we investigated the interactions with metals such as Zn^2+^ and Mn^2+^, for which we performed in silico docking on the MIB online server. We carry out the docking with Mn^2+^ because, in the superposition models, the thrown models present Mn^2+^ in their structure in addition to Zn^2+^. Furthermore, in the oligomerization experiments with the metals, we work with Mn^2+^ and not with Fe^2+^ due to the fact that iron oxidized more easily. The server locates these sites by predicting the metal ion’s binding site. The server generates templates with the different positions of the residues to which the metal can bind. In the case of Mn^2+^, nine templates were generated that contained nine amino acid residues that bind with this metal, and 22 were generated for Zn^2+^. Figure 3a shows one template structure of the MAP3773c protein containing Zn^2+^, and Figure 3b shows the Mn^2+^ ion-binding residues with the highest score, within a radius of and 2.756 Å for Zn^2+^ and 1.82 Å for Mn^2+^. 

The data obtained for MAP3773c were reported with the metals Zn^2+^ an Mn^2+^. The software reports the templates from the highest to the lowest score and gives them an ID/name; we replace it with number 1 for the first template with a higher score, and so on. Only the first template was presented for Zn^2+^ and Mn^2+^.

The binding distances between the metal ion and the amino acid are closely related to the differences in the binding potential [43]. In the first template for the case of Zn^2+^, the amino acids that interacted in all the templates were all cysteine (91, 94, 131, and 134), presenting distances of 2.375, 2.175, 2.672, and 3.38 Å from the metal ion, respectively. For the case of Mn^2+^, we found one template. Eight different amino acids interact in the first template, and those with a higher score for the interaction with the metal ions were E107 and H108, presenting distances of 3.414 Å and 4.566 Å, respectively.

### 3.3. Expression, Purification and Oligomerization of MAP3773c

The MAP3773c protein was the subject of our research due to its possible involvement in the pathogenicity of MAP, since it was hypothetically identified as a protein belonging to the family of FUR proteins, and it is involved in the capture of iron and zinc [4]. It also has the highest similarity and predicted structure resembling that of FurB-type protein [22], as we found in the crystal model of the protein PDB ID: 2O03 MTB. The overexpression of the protein MAP3773c in the strain of *E. coli* BL21 (DE3) was carried out under different induction conditions, and the optimal expression was found to be 16 h with 1 mM IPTG at 30 °C (data not presented). This report provides a reliable method for the efficient expression and purification of MAP3773c and growth conditions that may be useful for further studies. Efficient overexpression and purification of recombinant MAP3773c is a prerequisite for future functional and pathogenicity involvement studies. MAP3773c purification was performed in a single stage using a Zn-imido-acetate column. In our case, imidazole was the eluent of choice, and the pH at which MAP3773c presented a higher solubility was pH 4 in a 10 mM acetate/acetic acid buffer. Although the theoretical isoelectric point of the recombinant protein was 9.23, a working pH of 5 was chosen to be as close as possible to neutrality and to avoid oligomerization, which is useless for characterization and functionality analyses. The results of the SDS-PAGE electrophoresis for different aliquots taken throughout the MAP3773c protein purification procedure are shown in (Figure 4). Two forms or two bands of the protein appeared in the gel of approximately 20 kDa and 17 kDa, which are probably due to the different redox states of the cysteines [43,44]. The theoretical molecular weight of the protein is 16.22944 kDa, which corresponds to the translatable region of the map3773c gene. The figure is presented as Appendix A. The MAP3773c protein did not contain contaminating DNA, as observed in the gels stained with ethidium bromide (data not presented). 

The concentration of MAP3773c determined by the Bradford method was 1.15 mg/mL, in monomer equivalents for every 10 g of bacteria. Interestingly, the reaction of MAP3773c in the presence of Mn^2+^ or Zn^2+^ led to the formation of monomers and dimers. By means of EMSA, Shoyama et al. [23] verified the interaction of the MAP3773c protein with a region of 19 bp called the iron box–which is also located in *E. coli* [8]–under conditions with and without manganese; under increasing manganese conditions, the interaction was greater [23]. Regarding its oligomerization properties (Figure 5), the recombinant protein MAP3773c in the oligomeric state was studied by SDS-PAGE in the presence and absence of reducing agents. In Lane 2, under reducing conditions with β-mercaptoethanol, MAP3773c was reduced to two monomers of different molecular weights–one of approximately 15 kDa and the other of 20 kDa–with a fragment of 51 kDa and another of approximately 68 kDa. We consider that the 51 kDa fragment is possibly a trimer of 15 kDa monomers and the 68 kDa fragment is a tetramer of 17 kDa monomers. Therefore, the explanation we have in this regard is that MAP3773c underwent monomer lysis, forming a monomer doublet. In Lane 3, for the case with DTT, a monomer of approximately 17 kDa was identified. In Lane 4, under reducing conditions with DTT and β-mercaptoethanol, the behavior of MAP3773c was similar to that with β-mercaptoethanol (Lane 2). In Lane 6, under reducing conditions with β-mercaptoethanol and with Mn^2+^, MAP3773c presented the same oligomers as in Lane 2 with β-mercaptoethanol; possibly, Mn^2+^ does not reverse or prevent the action of reduction by β-mercaptoethanol. For the case of the reaction with Mn^2+^ only, without β-mercaptoethanol (Lane 7), bands of 15 and 100 kDa identical to those under non-reducing conditions were observed. In Lane 8, in the case of Zn^2+^, the protein fragments observed under reducing conditions were similar to those of Mn^2+^ under reducing conditions, with the exception of a fragment of approximately 60 kDa, which is a trimer. This fragment did not appear under any other treatment, which indicates that this trimer was formed by the action of Zn^2+^. We also observed that under these conditions, a 100 kDa pentamer was formed, which appears only under non-reducing conditions. In Lane 9, like the other reactions without β-mercaptoethanol, a monomer of approximately 15 kDa was observed and the pentamer that was formed in other reactions under non-reducing conditions (Lanes 3, 5 and 7) did not form. Therefore, according to our results, we believe that MAP3773c forms oligomers under reducing conditions with Zn^2+^, but not with Zn^2+^ alone. 

However, these bands could possibly be from some artifact of contaminant, and until we carry out a Western blotting, we cannot ensure such a behavior of the protein before the reducing agent and zinc. Therefore, as an alternative interpretation of the results shown in Figure 5, the effect of zinc on oligomerization is that the MAP3773c trimer was stabilized by Zn^2+^ against SDS denaturation and β-mercaptoethanol reduction, as is the case for other proteins such as PerR from *Bacillus subtilis*. When treated with DTT as the reducing agent, the zinc remained bound [34]. This effect of zinc on oligomerization was evident, so we can also say that in MAP3773c, as in Zur from *E.coli*, metal binding changes the high-affinity conformational equilibrium [45]. This suggests that the presence of a reducing agent and zinc is necessary for the formation of oligomers. However, in the case of the reaction with zinc and MAP3773c, the formation of oligomers was not dependent on the DNA, and the protein can form oligomers only with zinc and manganese. This feature was not found for all Zur proteins; for example, in the case of *Streptomyces*, Zur did not form oligomers in the absence of DNA [46]. The presence of dimer and oligomer in highly purified FUR proteins is common [46,47].

### 3.4. Circular Dichroism, Crosslinking, and PAR Staining

The spectrum of CD generated by MAP3773c (Figure 6a) presents many alpha helices observed at an approximate absorbance of 190 nm, which is characteristic of this structure [48]. These data are consistent with those obtained with the DichroWeb Server report given in Table 1. The results of the circular dichroism performed on the server revealed a high prevalence of alpha helices in the secondary structure (50%) and lower proportions of β-sheets (20%), turns (7%), and disordered regions (23%), as presented in Figure 6b. This agrees with previous structural data on Fur proteins–found in other bacterial species–and in Zur-type proteins [49], thus indicating that the state confirmed by the CD is the correct folding. Our results are supported by work carried out on ⍺-helical proteins, in which the authors mention that ⍺-helical proteins have a positive band at 193 nm and negative bands at 222 nm and 208 nm [50]. 

We suggest that the MAP3773c protein contains structural zinc (Figure 7a), according to the results of PAR staining and SDS-PAGE electrophoresis, as well as the docking results, which showed the affinity of MAP3773c for zinc. In the absence of β-mercaptoethanol, no zinc band was observed; however, with β-mercaptoethanol, structural zinc was detected, which allowed the zinc to be released by H_2_O_2_ and a thiol-oxidizing agent [35], and detected with the PAR reagent. In (Figure 7a) in Lane 2 (MAP3773c without mercaptoethanol), zinc was not detected, while in Lane 3 (MAP3773c with mercaptoethanol), zinc was detected in a 100 kDa oligomer, which was apparently a pentamer. In Lane 4 with DTT, zinc was also detected as a trimer; finally, in Lane 5 (MAP3773c with H_2_O_2_), no zinc was detected. Our results show (Figure 7b) that the MAP3773c monomers of 15 and 20 kDa do not contain zinc due to the action of the reducing agent Zn^2+^ leaving the molecule, which is why it was not detected with PAR staining as in other Zn-containing proteins. For example, in PerR from *B. subtilis*, zinc was detected in the protein when treated with DTT and under non-reducing conditions; in this case, zinc could be detected in a monomer of approximately 14.4 kDa and not in the 16.4 kDa monomer, but not under oxidizing conditions with H_2_O_2_ [35]. However, in MAP3773c, zinc could be detected in a pentamer of 100 kDa, which we believe to be a MAP3773c protein pentamer formed by the reducing agent in the case of β-mercaptoethanol and with DTT. Oligomers of a molecular weight lower than that were not clearly differentiated. Through the action of the reducing agent, zinc is trapped in the structure of the oligomer; later, through the action of H_2_O_2_, the Zn comes out and can be detected with the PAR dye.

### 3.5. Western Blotting Analysis of Map3773c

The Western blotting analysis (Figure 8) showed that the protein was recognized by anti FurB antibodies from *Anabaena* sp. PCC7120, but not for the antibodies anti FurA. FurB from *Anabaena* sp. PCC7120 functions by regulating the genes implicated in zinc metabolism and those controlling the oxidative stress response. Western blot results indicate that the antigenic determinants of MAP3773c are more similar to those in FurB suggesting the MAP3773c structure, and therefore its function could be more related to FurB than to FurA from *Anabaena*. Considering the recognition of MAP3773c by the antibodies raised against FurB from *Anabaena* sp. PCC 7120, we performed an alignment to determine the similarity of the two proteins (Figure 9). The alignment showed a similarity of 28%.

### 3.6. BN-SDS PAGE and BN PAGE Immunoblotting

BN-PAGE (Figure 10) allows the analysis of the aggregation or oligomeric states of proteins in cell extracts. To analyze whether the MAP3773c protein formed multiprotein complexes, i.e., dimers, trimers, etc., we first separated the protein in 2D under native conditions (Figure 10a). Furthermore, we detected the protein using an anti-His-tag antibody (Figure 10b). MAP3773c was in the form of a monomer in the native MP extract, and we could determine this in the 2D BN-PAGE of the subcomponents with Coomassie Brilliant Blue and with the anti-His-tag antibody. The BN-PAGE immunoblotting only revealed the monomer form. Thus, we demonstrated that our protein, under native conditions, is present as a monomer without forming oligomers.

### 3.7. LC-MS of the Purified Monomer of MAP3773c

The purification of MAP3773c only revealed the monomer form when the mass was determined by LC-MS. As it can be seen in (Figure 11), only one form (monomer) was present in the sample of the purified protein. Our results determined that nearly 100% of our sample comprised MAP3773c, thus confirming the purity of the protein and that the eluted MAP3773c has a molecular weight of approximately 20,564.31 Da. Given that SDS-PAGE cannot be used to determine the exact mass of the protein, LC-MS was used to confirm the exact molecular weight of MAP3773c, which was estimated to be 20,569.69 Da and is in good agreement with the theoretical molecular weight determined by the sequencing of the cloned gene in pRSET-A.

### 3.8. Oligomeric State of the Fur Protein, MALDI-TOF MS

In the (Figure 12) MALDI-TOF MS, no oligomeric forms were detected and a monomeric form with a predominant mass of 20,975 *m*/*z* was observed (and another of 9871that is presumed to be the middle ion). However, the signal is weak since 1000 units are not reached for the mass of 20,975 *m*/*z*. The spectrum is cut off near 10,000 *m*/*z*, so it is unknown whether there are contaminants smaller than this mass that are presumed to have been removed during protein washes using acetate buffer and a 10,000 MW amicon. The molecular weight of MAP3773c by the LC-MS technique was 20,564.; this molecular weight corresponds to the molecular weight of the MAP3773c protein, a segment of the Xpress protein and another segment of the histidine tag (6 his-tag). Similary, the molecular weight obtained by MALDI-TOF MS technique was 20,975; the molecular weight has a difference of 410Daltons, which is possibly due to the effect of the matrix used for the MALDI-TOF MS analysis. In the case of LC-MS, it is a direct measurement of the sample without a prior separation method. Another possible cause is that the samples for both analyzes were purifications that were performed at different times and the sample that was used for MALDI-TOF MS was possibly contaminated with DNA, which possibly caused the molecular weight of MAP3773c to increase [51].

## 4. Conclusions

In the present study, we provided a detailed analysis of new findings on the MAP3773c protein through an in silico modeling study and in vitro studies of some of the biochemical characteristics and the structural zinc. We demonstrated both by in silico analysis and experimentally that MAP3773c contains structural zinc. According to the structural models and the percentage of similarity, which was 34% for the protein PDB ID: 2O03 of *M. tuberculosis*, zinc could be coordinated by the two canonical CXXC motifs involving cysteines 91, 94, 131, and 134. On the other hand, MAP3773c was also recognized by the anti-FurB antibodies from *Anabaena* sp. PCC7120, but not by anti-FurA antibodies. Our results showed that MAP3773c is a Zur-type protein that is a monomer in vivo. In this study, we have provided tools to prepare MAP3773c in a pure form using a single step of purification, as well as the optimal conditions for the growth, expression, and higher yield of the protein MAP3773c. The use of this purification method allowed us to obtain the monomer form. We demonstrated that MAP3773c protein has a monomer form under native conditions, which was verified by performing a MALDI-TOF MS analysis, which presented only the monomer form.

## Figures and Tables

**Figure 1 biology-11-01183-f001:**
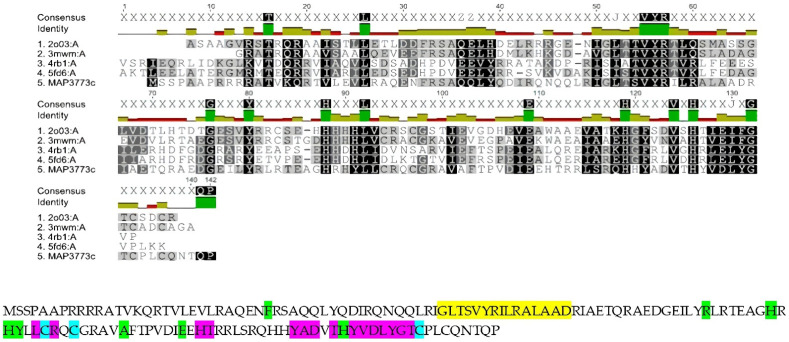
Sequence alignment of MAP3773c with FUR proteins with available structures that show the highest similarity and down, conserved protein domain, yellow: putative DNA binding helix, green: metal binding site 1 and 2, blue: structural Zn^2+^ binding site, magenta: polypeptide binding site.

**Figure 2 biology-11-01183-f002:**
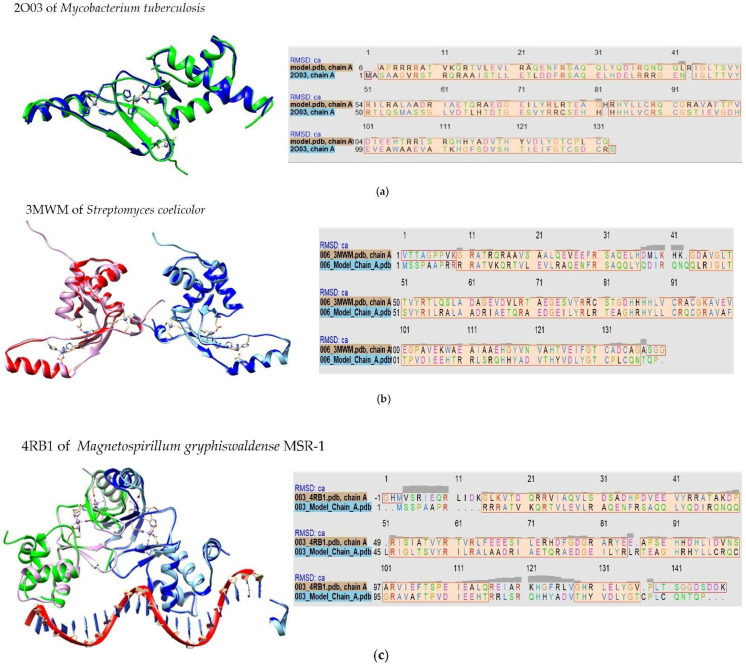
In silico models of the MAP3773c protein of MAP, PDB ID: 2O03 of *M. tuberculosis*, PDB ID: 4RB1 of *M. gryphiswaldense*, PDB ID: 3MWM of *S. coelicolor* and PDB ID: 5FD6 of *R. leguminosarum*. (**a**). Superposition of the blue crystals of protein PDB ID: 2O03 and the model built for MAP3773c in green. (**b**) Overlay between the model (light blue and purple) and the PDB ID: 3MWM crystal (dark blue and red), showing the alignment between Chord A of the model (bottom, in blue) and Chord A of the crystal (top, PDB ID: 3MWM). (**c**) Overlay between the model (light blue, Chain A; magenta, Chain B) and the PDB ID: 4RB1 crystal (dark blue, Chain A; green, Chain B). (**d**) Overlay between the model (pastel colors) and the crystal structure (strong colors). Alignment between Chord A of the model (bottom, in blue) and Chord A of the crystal (top, PDB ID: 5FD6). The bars labeled with the RMSD show where the greatest difference occurred between the crystal and the model, showing the greatest difference between the handles.

**Figure 3 biology-11-01183-f003:**
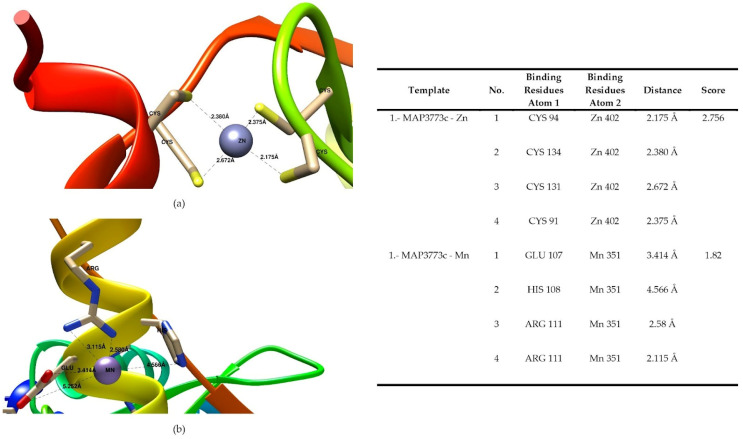
Docking presentation of the binding sites for metal ions (Mn^2+^ and Zn^2+^) in MAP3773c. Template of the MAP3773c model with (**a**) the Zn^2+^ ion. (**b**) with Mn^2+^.

**Figure 4 biology-11-01183-f004:**
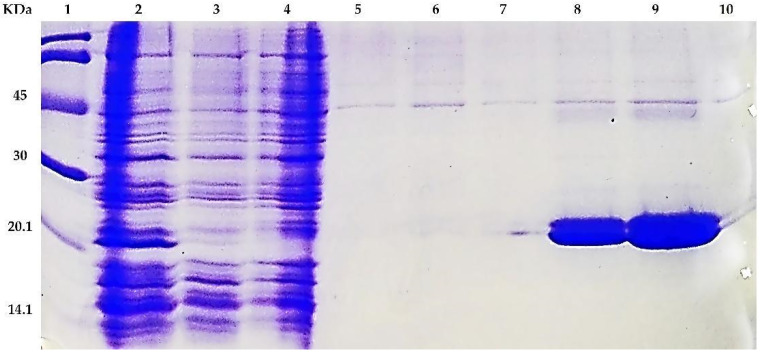
Expression and purification of MAP3773c in *E. coli*. SDS-PAGE electrophoresis of the purified MAP3773c recombinant protein. Lane 1, molecular weight marker; Lane 2, filtered extract; Lane 3, filtered extract passed through the column; Lane 4, Wash 1; Lane 5, Wash 2; Lane 6, Elution 1; Lane 7, Elution 6; Lane 8, Elution 11; Lane 9, Elution 16; Lane 10, Elution 21.

**Figure 5 biology-11-01183-f005:**
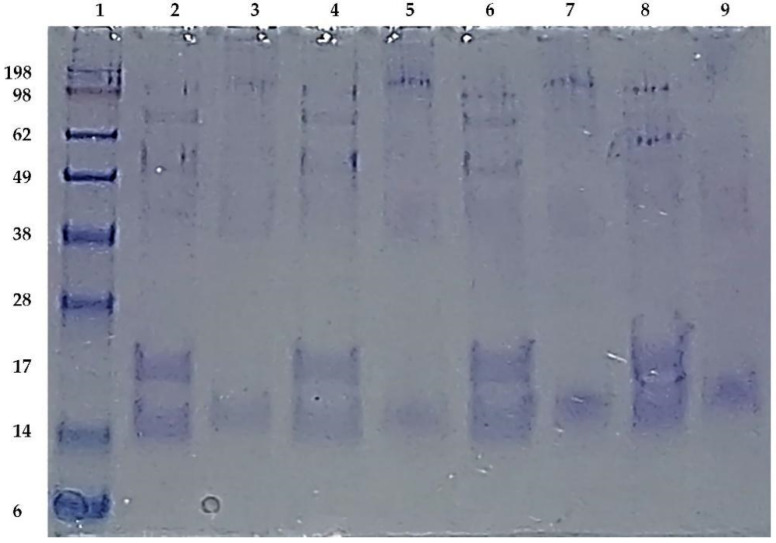
SDS-PAGE electrophoresis of MAP3773c showing the oligomerization of MAP3773c under different conditions. Lane 1, molecular weight marker; Lane 2, MAP3773c with β-mercaptoethanol; Lane 3, MAP3773c without β-mercaptoethanol; Lane 4, MAP3773c with 10 mM DTT and β-mercaptoethanol; Lane 5, MAP3773c with DTT and without β-mercaptoethanol; Lane 6, MAP3773c with 15 mM Mn^2+^ with β-mercaptoethanol; Lane 7, MAP3773c with 15 mM Mn^2+^ without β-mercaptoethanol; Lane 8, MAP3773c with 15 mM Zn^2+^ with β-mercaptoethanol; Lane 9, MAP3773c with 15 mM Zn^2+^ without βmercaptoethanol.

**Figure 6 biology-11-01183-f006:**
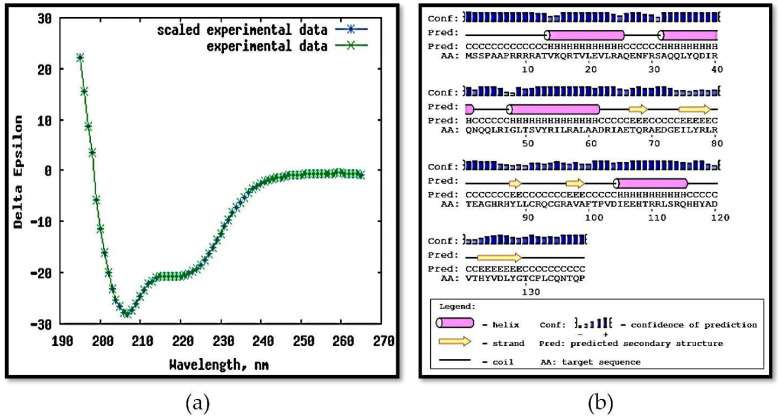
Secondary structure studies of MAP3773c. (**a**) Spectrum generated by circular dichroism analysis of MAP3773c recombinant protein from 195 nm to 265 nm. (**b**) Secondary structure prediction from MAP3773c protein. Data were obtained using the PSIPRED web server (http://bioinf.cs.ucl.ac.uk/psipred/, accessed on 12 July 2016).

**Figure 7 biology-11-01183-f007:**
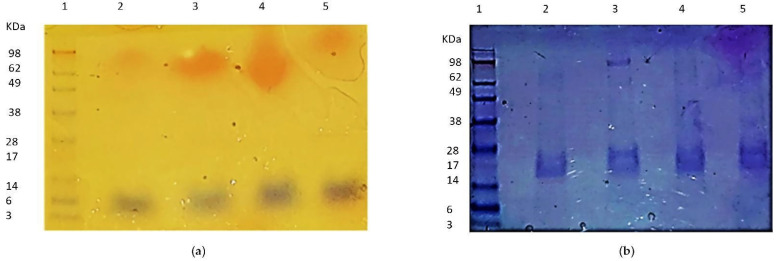
Electrophoresis SDS-PAGE for testing of presence of structural zinc of recombinant MAP3773c by stain PAR. (**a**) Lane 1: molecular weight marker; lane 2: MAP3773c without β-mercaptoethanol; lane 3: MAP3773c with β-mercaptoethanol 4, MAP3773c with DTT; lane 5: MAP3773c with H_2_O_2_. (**b**) The same gel as A, stained with Coomassie blue.

**Figure 8 biology-11-01183-f008:**
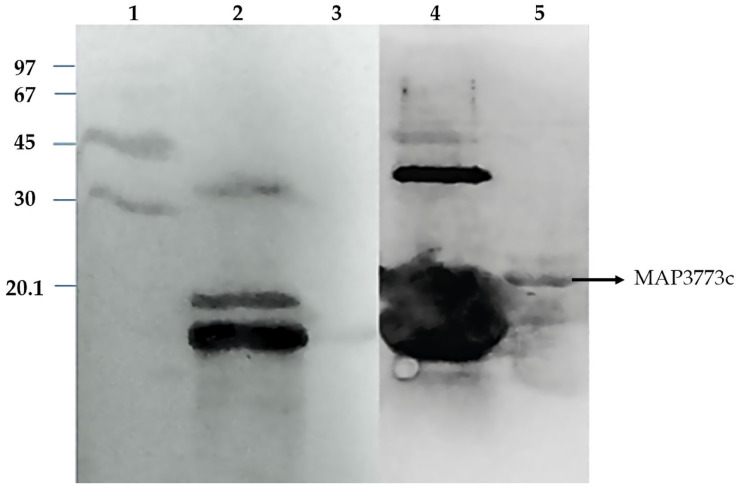
Western blot analysis of MAP3773c tested with antibodies raised against recombinant FurA and FurB from *Anabaena* sp. PCC7120. Line 1, MWM; Line 2, Western blot of FurA from *Anabaena* sp. PCC7120 protein tested with antibodies raised against FurA; Line 3, MAP3773c tested with antibodies anti-FurA from *Anabaena* sp. PCC7120; Line 4, FurB from *Anabaena* sp. PCC7120 tested with anti-FurB antibodies; Line 5, MAP3773c tested with *Anabaena* sp. PCC7120 anti FurB antibodies.

**Figure 9 biology-11-01183-f009:**
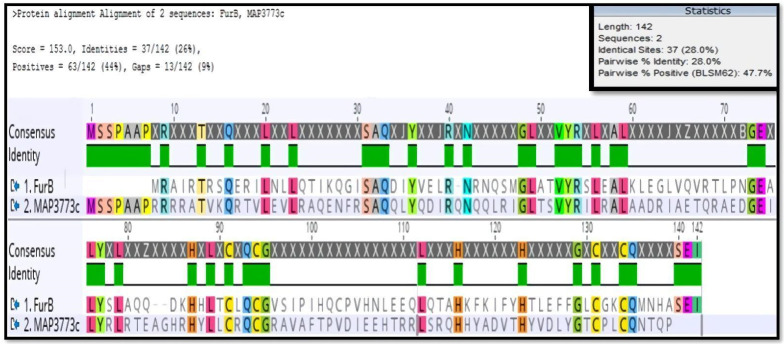
Bioinformatic alignment of the amino acid sequences of the MAP3773c protein of MAP and FurB of *Anabaena* sp. PCC7120. WP_010996629.

**Figure 10 biology-11-01183-f010:**
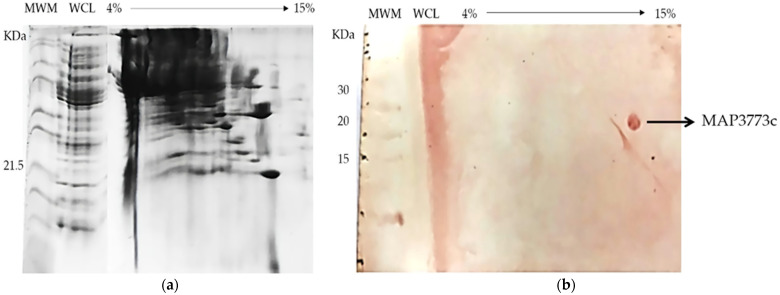
In (**a**), BN-PAGE of extract protein cellular lysates from *E. coli* host cell expressing recombinant MAP3773c, followed by (**b**) an immunoblot assay to separate possible complexes representing MAP3773c. WCL (whole-cell lysate).

**Figure 11 biology-11-01183-f011:**
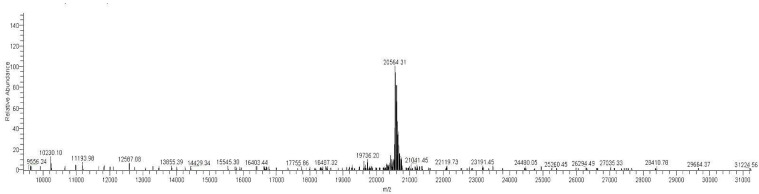
Molecular mass of MAP3773c. Mass spectrometry of MAP3773c with the monomeric form, using LC-MS. Total ion count LC-MS for MAP3773c demonstrated one peak. This peak represents a single, charged MAP3773c with an average mass of 20,564. Da.

**Figure 12 biology-11-01183-f012:**
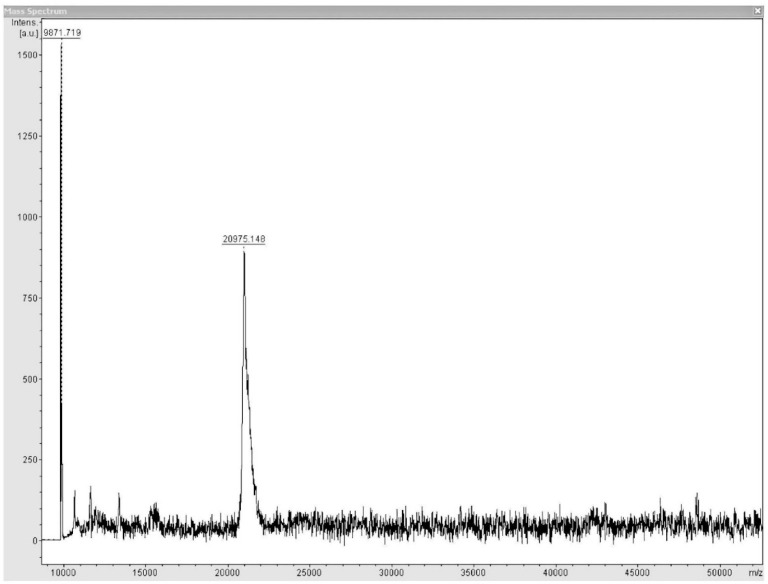
MALDI-TOF MS of the MAP373c protein, shown in monomeric form.

**Table 1 biology-11-01183-t001:** Interpretation of the circular dichroism analysis performed on the recombinant MAP3773c protein.

Secondary Structure of the Recombinant Protein MAP3773c			
Alpha helix	Folded Sheet-Beta	Twists	Disordered
50%	20%	7%	23%

## Data Availability

Not applicable.

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
