# Peer review of "In Silico and In Vitro Analysis of MAP3773c Protein from Mycobacterium avium subsp. Paratuberculosis"

_biology, 2022, doi:10.3390/biology11081183_

Round 1

Reviewer 1 Report

This study reports the investigation of MAP3773c protein using in silico and in vitro methods to analyze some of the biochemical characteristics and the structural zinc. Here, the presented data suggest MAP3773c is a Zur-type protein and under native conditions is a monomer. The following concerns should be addressed prior to the publication of the paper.

-Sec. 2.1: Please clarify the selection of structural templates for homology modeling.

-Sec. 3.1: Please clarify the rationale for the selection of 4 proteins for in silico modeling.

-Figure 3: Please consider substituting the generic software-generated tables. The unit of score values is missing in the table as well.

-Sec. 3.7: Please consider including the theoretical molecular weight determination data in a supplementary information document.

Reviewer 2 Report

I haven't read such a well-structured manuscript in ages where all the essays talk well to each other. 

The present manuscript has not skimped on essays to reach its goals. 

However, I strongly recommend that the authors perform a molecular dynamics simulation of the homologous model of at least 100ns. Importantly. In my experience, this always allows refining the model structure and brings more reliability in the use of the model.

I would like to have seen any assays of candidate MAP3773c inhibitors that have been selected by docking analyses and had their activities evaluated in vitro, with the purified target.

Another point: why not use a MALDI-TOF-MS to confirm the band of the protein? I believe it is a more robust and reliable technique for this type of analysis.

Reviewer 3 Report

Review of Guevara et al. for MDPI Biology

In their manuscript entitled, “In Silico and In Vitro Analysis of Map3773c Protein from Mycobacterium avium subsp. paratuberculosis,’’ Guevara et al. presented the cloning of the map3773c gene from Mycobacterium avium and its plasmid-based inducible expression in an E. coli expression host.  The 6XHis-tagged recombinant MAP3773c was isolated and enriched using a single-step metal-affinity purification. Successful isolation of the recombinant MAP3773c was confirmed by Coomassie Blue staining, Western blotting, and mass spectrometry. Because the crystal or NMR structure of MAP3773c is not available in the Protein Data Bank, the authors used in silico approaches and circular dichroism to predict and elucidate, respectively, the secondary structures of MAP3773c. Its tertiary structure was modelled using the atomic coordinates of the X-ray crystal structures of Fur B proteins from M. tuberculosis, Streptomyces coelicolor, Magnetospirillum gryphiswaldense, and Rhizobium leguminosarum. Through alignment of conserved residues that previously determined to coordinate metal ions in Fur B homologs, coupled with metal-binding prediction algorithms, the authors suggested that MAP3773c coordinates Zn2+ and, probably, Mn2+. In addition, the authors also claimed that MAP3773c form homotrimeric or homopentameric isoforms and coordinate Zn2+ only during reducing conditions.

Although this study contributes some insights into the structure of MAP3773c, there are some weaknesses that should be addressed prior to publication.

Major Points:

1.    The authors failed to provide sufficient evidence supporting that the isolated recombinant MAP3773c is homogeneously pure. Single-step metal-affinity purification alone (not in tandem with size-exclusion chromatography, ion-exchange chromatography, or immuno-capture precipitation as a secondary purification step), usually yields impure proteins even when stringent imidazole washing is used. High-MW contaminants are apparently observed in Coomassie Blue gels (Fig. 4, Fig. 5, and Fig. 7B). Staining with a more sensitive stain, such as Silver Stain or SyproRuby, would give us a better picture whether MAP3773c was purified to homogeneity. Other peaks showing up in the LC-MS spectra is also indicative of impurity (Fig. 11). The absence of co-eluted DNA was determined only by ethidium bromide staining, which is not reliable in detecting variable sizes of contaminating E. coli genomic DNA. Alternatively, supplementation with DNAse prior to centrifugation would help in degrading DNA in the cell lysate.

2.    Throughout the manuscript the authors claimed that the purified recombinant MAP3773c is a monomer

****Please see attachment***

Round 2

Reviewer 1 Report

Sec 2.1 and 3.1: Please consider substituting “PDB 2O03, PDB 3MWM, PDB 4RB1 and PDB 5FD6” with “PDB IDs: 2O03, 3MWM, 4RB1 and 5FD6”. Also please consider substituting “PDB” with “PDB ID” (typical comment for entire paper).

Sec 2.2: Please include the URL in “()” and consider adding the server’s name similar to the supplementary material.  

Figure 3: If values in the score column have energy unit, please add it as well. Please consider substituting the template names of “1dvpA1” and “1ipsA1” if possible (for example it is not clear what A1 signifies).

Reviewer 3 Report

The authors addressed most of the corrections by incorporating them into the revised version of the manuscript.  However, the authors are still overstating their conclusions on Fig. 5, of which the MAP3773c can exhibit a pentameric structure primarily based on a faint ~100 kDa band that appears under reducing condition without even confirming it by Western Blot. Such bold statement, however, is not bolstered by sufficient convincing evidence. This high-MW band can be a contaminant or a non-physiological artifact due to covalent crosslinking. In addition, although it is stated that the theoretical molecular weight of the protein is 16.22944 kDa, it is unclear in the manuscript whether this mass is the endogenous MAP3773c or the recombinant variant (including the affinity tag and amino acid linkers; actual protein translation of the plasmid's coding region). The monoisotopic and average masses of the recombinant MAP3773c should be exact values and not estimates.  It is also confusing why the major peak at 20975.148 m/z as detected by MALDI-TOF is hundreds of Daltons off compared to the theoretical mass and observed mass (by LC-MS). 

Round 3

Reviewer 3 Report

The current revised version of the manuscript is acceptable for publication.

Author Response

Thank you very much for your review, for your suggestions, for all the work I do for our manuscript to finally be publishable. We thank you for all the learning obtained, we learned a lot, thank you very much. We hope that in a future let us meet again for a next review of the next manuscript. We will make the English language corrections and we are willing to cover any expense if the Publisher makes any corrections to the language.